# Unemployment Trajectories and the Early Risk of Disability Pension among Young People with and without Autism Spectrum Disorder: A Nationwide Study in Sweden

**DOI:** 10.3390/ijerph17072486

**Published:** 2020-04-05

**Authors:** Tea Lallukka, Ellenor Mittendorfer-Rutz, Jenni Ervasti, Kristina Alexanderson, Marianna Virtanen

**Affiliations:** 1Department of Public Health, University of Helsinki, P.O.B. 20, 00014 Helsinki, Finland; 2Department of Clinical Neuroscience, Division of Insurance Medicine, Karolinska Institutet, SE-171 77 Stockholm, Sweden; ellenor.mittendorfer-rutz@ki.se (E.M.-R.); kristina.alexanderson@ki.se (K.A.); marianna.virtanen@uef.fi (M.V.); 3Finnish Institute of Occupational Health, Work Ability and Working Careers, P.O.B. 18, 00032 Helsinki, Finland; jenni.ervasti@ttl.fi; 4School of Educational Sciences and Psychology, University of Eastern Finland, P.O.B. 111, 80101 Joensuu, Finland

**Keywords:** autism spectrum disorder, risk factor epidemiology, population-based, trajectory analysis, person-oriented methods, unemployment, disability pension, social determinants

## Abstract

Depression and anxiety are associated with unemployment and disability pension, while autism spectrum disorder (ASD) is less studied. We aimed to first identify unemployment trajectories among young adults with and without ASD, and then to examine their social determinants. Finally, we used the trajectories as determinants for subsequent disability pension. We used a population-based cohort, including 814 people who were 19–35 years old, not on disability pension, and who had their ASD diagnosis between 2001 and 2009. A matched reference population included 22,013 people with no record of mental disorders. Unemployment follow-up was the inclusion year and four years after. Disability pension follow-up started after the unemployment follow-up and continued through 2013. We identified three distinctive trajectories of unemployment during the follow-up: (1) low, then sharply increasing (9%,) (2) low (reference, 67%), and (3) high then slowly decreasing (24%). People with ASD had higher odds of belonging belong to the trajectory groups 1 (OR 2.53, 95% CI 2.02–3.18) and 3 (OR 3.60, 95% CI 3.08–4.19). However, the mean number of unemployment days was relatively low in all groups. A disability pension was a rare event in the cohort, although memberships to groups 1 and 3 were associated with the risk of a future disability pension. More knowledge is needed about factors facilitating participation in paid employment among people with ASD.

## 1. Introduction

Unemployment is a notable challenge, and it is important to identify unemployment trajectories and their determinants among young people, to be able to support their participation in paid work. Unemployment is strongly associated with mental disorders, disability pension (DP) and even suicide, with a worsening prognosis the longer the duration of unemployment continues [1,2,3,4,5]. There is, however, relatively little research on early onset mental disorders and their contribution to unemployment and the early risk of permanent exit from paid employment, in terms of disability pension. Such research is largely focused on the most prevalent mental disorders including depression and anxiety [6], while much less is known about other mental disorders such as neurodevelopmental disorders including autism spectrum disorder (ASD) regarding these outcomes. Studying unemployment trajectories is evidently only relevant to people with or without ASD who are not receiving a disability pension and are at risk of unemployment and future enrolment in a disability pension. It is known that a weak attachment to the labour market in terms of e.g., unemployment, DP, and early old-age retirement, is common among people with ASD [7,8]. Based on a recent review, people with ASD, who are in paid work may also face more challenges and e.g., poorer both physical and mental health, as compared to other workers [9].

Another review [10] highlighted the inconsistency of previous evidence regarding good outcomes in adult life among people with ASD, e.g., with respect to jobs, and how social factors contribute to the outcomes. Thus, such factors should be considered to increase understanding about the distinct developmental patterns in unemployment trajectories, alongside ASD. Additionally, an important aspect is the influence of the severity of the ASD, and when studying (un)employment, only “high functioning” people with ASD have typically been included [9,11]. “Low functioning” ASD means that ASD is co-occurring with intellectual disability, and although the patterns of ability are wide, the groups (“high functioning” and “low functioning”) differ in terms of their subsequent health outcomes [12].

Finally, to better manage heterogeneity in the population, and identify distinct groups of people who are likely to follow similar developmental trajectories in the course of their unemployment during the follow-up, person-oriented methods are needed [13]. In the variable-oriented methods, instead of the focus on the relationships among individuals with different unemployment patterns, the focus would be on the relations between variables [14], such as between mental health and subsequent unemployment. The variable-oriented approach thus assumes that all people in the cohort form a homogenous group with a similar risk of future unemployment and enrolment in a disability pension. In the person-oriented approach, trajectories arise from the data, to approximate actual developmental patterns. Furthermore, these patterns (trajectories) cannot be directly observed from the data, e.g., using a series of repeated measurements of mean development in unemployment days. In other words, with the method, we can identify latent groups of individuals in the data, without making any assumptions a priori. As it is unlikely that all participants’ development is close to any average trajectory, by assuming such homogeneity, we would miss these distinct developmental patterns, and the opportunity to identify new risk groups, or groups of people with more optimal development. Moreover, after identifying these latent groups, it is possible to study their determinants, and also use the latent group memberships to study their consequences for subsequent outcomes (here: disability pension).

In sum, despite the above described and overall poorer outcomes among people with ASD as compared to a same-aged reference population [10], many people with ASD nonetheless are active in the labour market, making it an important question to address how ASD is related to future unemployment trajectories, and how such distinct trajectories are associated with future disability pension. Thus, to be able to promote sustained employment, a more comprehensive understanding of unemployment trajectories, and their contribution to a subsequent permanent exit from paid employment is needed. 

The aims of the study were as follows:To identify unemployment trajectories among young adults (aged 19–35), and to examine ASD and social factors as determinants of the future unemployment trajectory memberships.To estimate mean number of unemployment days per year among people with ASD, their references, and within the distinct trajectory groups.To use the identified trajectories as determinants of the subsequent risk of disability pension, considering the independent effects of ASD.

## 2. Materials and Methods 

A prospective cohort study was conducted. The data for this study were obtained from the population-based nationwide registers in Sweden (Figure 1). We used the Statistics Sweden’s Longitudinal Integration Database for Health Insurance and Labour Market Studies (LISA) for information on sex, age, education, birth country, type of living area, and number of unemployment days [15]. The National Board of Health and Welfare’s registers comprise data from the National Patient Register including diagnosis-specific information on inpatient and specialized outpatient healthcare (the International Classification of Diseases, ICD-10) [16] (diagnoses and dates) and the Cause of Death Register for dates of deaths. The National Social Insurance Agency’s registers (MIDAS) were used to obtain information on disability pension dates (ICD-10). Ethical approval for the project was granted by the Regional Ethical Review Board of Stockholm, Sweden (DNR: 2007/5:6, 2016/1533-32). For a fully register-based study, no informed consent was required. The data used in this study are administered by the Division of Insurance Medicine, Karolinska Institutet, and cannot be made publicly available. According to the General Data Protection Regulation, the Swedish law SFS 2018:218, the Swedish Data Protection Act, the Swedish Ethical Review Act, and the Public Access to Information and Secrecy Act, these types of sensitive data can only be made available from the respective authority, after legal review, for researchers who meet the criteria for access to this type of sensitive and confidential data. Readers may contact Professor Kristina Alexanderson regarding the data. 

All people aged 19–64 and living in Sweden can be granted a disability pension (DP) on a full- or part-time basis if they have permanently reduced their work capacity due to a disease or injury. Young adults (19–29 years) can be granted a temporary DP for having long-term reduced work capacity due to morbidity or difficulties in completing education in the required time [17]. People can receive unemployment benefits when actively seeking paid work.

### 2.1. Individuals with Autism Spectrum Disorder and their Matched References from the General Population

ASD was indicated based on the first date of recorded ICD-10 codes available in inpatient and specialized outpatient health care during the period 2001 to 2010. ASD was defined as: F84.0. Childhood autism; F84.1. Atypical autism; F84.3. Other childhood disintegrative disorders; F84.5. Asperger’s syndrome; F84.8. Other pervasive developmental disorders; F84.9. Pervasive developmental disorder, unspecified. These comprehensive data cover all people diagnosed during the inclusion years, however, not those who either were not diagnosed or were only diagnosed or were only diagnosed and treated in primary healthcare (the latter is very unlikely). We included all people who were 10–35 years old, when their ASD was first recorded. For those diagnosed before age 19, the follow-up for unemployment began the year when they turned 19. Moreover, all had to have lived in Sweden at least five years before their ASD diagnosis was recorded. For each individual with ASD, we included five randomly selected matched references from the general population. These references were matched based on their sex, age, living area, and country of birth. As many were diagnosed with ASD before finishing school or entering paid work, matching on an educational level was not feasible. To be included as a reference, the individuals were also required to have been living in Sweden for at least five years before their cohort entry, and to have no indication of any mental disorders (ICD-10 codes F00-F99) in the available patient registers throughout the entire follow-up period (from 1987 to 2013).

People already on a DP at inclusion were excluded (Figure 1), as they could not have any unemployment benefits. Moreover, the subsequent risk of being on a DP can only be studied among people who were not on a DP at the start of follow-up for DP. This is the reason those granted a DP in the first five-year follow-up were excluded. Following a previous study [12], the analyses were conducted among people with ASD without intellectual disability (“high functioning”). There were only 13 “low functioning” ASD individuals in the final data, as defined based on their diagnosis codes regarding the intellectual disability of different levels of severity (F70 Mild mental retardation; F71 Moderate mental retardation; F72 Severe mental retardation; F73 Profound mental retardation; F78 Other mental retardation; F79 Unspecified mental retardation). This is because most people with “low-functioning” ASD were on full-time DP throughout the follow-up, and hence not eligible to be included in this study.

### 2.2. Unemployment Trajectories and Disability Pension Follow-Up

The follow-up for unemployment was restricted to five years (the inclusion year and four years after) for all participants in order to have the same number of time points for more reliable trajectories. All those who died, emigrated or had a disability pension event during the unemployment follow-up were excluded from the trajectory analyses, as they were not at risk of unemployment during the entire follow-up. Thus, using cohort entry dates, we only included people with an index diagnosis date in 2001–2008.

The unemployment variable was calculated for all cohort members for each follow-up calendar year, based on the number of days with unemployment benefits during that year of follow-up (zero days = 0, one day or more = 1). Additionally, mean unemployment days per year were examined among people with ASD, their references, and by the trajectory groups. The year zero reflected unemployment during the year of the diagnosis and the matched year for the references, while years 1 to 4 reflected years after the diagnosis year.

After the unemployment follow-up had ended, we followed the participants through 2013 to assess the risk of a disability pension. The follow-up ended at the date of disability pension, death, emigration or 31 December 2013, whichever occurred first. Mean follow-up time was 3.2 years (standard error 0.01).

Data formation, inclusion and exclusion criteria are further displayed in Figure 1. 

### 2.3. Covariates

As covariates, we used sex, continuous age, educational level (classified into low (elementary school), intermediate (secondary school), and high (college or university)), birth country (Sweden vs. other, due to a higher prevalence of ASD found among children born to immigrant parents compared to native families [18]), and type of living area (big cities, medium-sized cities, and rural areas). Initially, marital status was also considered (married or single), but due to the young age at cohort entry for most participants, the covariate was not very relevant, and it was omitted from subsequent analyses. Classifications further broadly follow our previous procedures [19,20,21].

### 2.4. Statistical Analyses

We identified trajectories of unemployment using a group-based trajectory analysis (GBTA). We selected the best model by testing different shapes of the trajectories and the ideal number of the groups, and compared Bayesian Information Criteria (BIC) values between more simple (less trajectory groups) and more complex (more trajectory groups) models (Appendix A). We also assessed mean posterior probabilities of the trajectory group membership, to confirm the reliability (accuracy) of the classification. In brief, the uncertainty related to group memberships can be quantified in the form of probabilities [22]. As all means were clearly above a recommended cut-off of 0.70, this supports the choice of the model. Moreover, the numbers of those with a posterior probability less than 0.70 were low in each trajectory group of the selected model. Finally, each group had to be reasonably sized for meaningful analyses and implications (group sizes close to 10% or more). All participants were assigned to the trajectory group for which they had the highest probability of group membership. The reliability of the classifications was also examined and found to be good or very good (posterior probability means from 0.79–0.96).

Next, we studied social and health-related determinants of trajectory group memberships. First, descriptively using chi-squared tests (cross-tabulations). Second, we fitted multinomial logistic regression models using trajectory membership as the outcome. 

Finally, the identified unemployment trajectories over five years were used to predict the risk of subsequent all-cause DP risk after the unemployment follow-up. Model 1 was adjusted for sex and age, Model 2 was adjusted for sex, age, and ASD and Model 3 for sex, age, ASD, birth country, living area, and educational level. Attenuation of the parameter estimates was computed with the formula comparing Model 1 to Model 0, and Model 2 to Model 1 [100*(Beta_Model 0 – Beta_Model 1)/(Beta_Model 0)]. We also tested whether there was any indication for Cox proportional hazards violation. The proportionality test suggested that there was no violation against the assumptions (p-value for the overall test checking the proportionality assumption was 0.227). All the analyses were performed using the SAS 9.4 Statistical Package (SAS Institute, Inc, Cary, NC). 

## 3. Results

### 3.1. Descriptive

The mean age among people with ASD was 22.6 years (95% CI 22.3-22.9, SD 4.97 years) at diagnosis, and 22.6 years among their references from the general population (95% CI 22.5-22.6, SD 4.77 years) at inclusion. Other descriptive characteristics of the study population are presented in Table 1. The proportion of men with ASD was higher than that of women. Of the people with ASD, less than 5% were married, and about 10% among their references. More than 90% were born in Sweden. There were no notable differences regarding the type of the living area.

### 3.2. Trajectories of Unemployment among People with and without Autism Spectrum Disorder

We identified three distinctive trajectories of unemployment among young people included in our analytical sample (Figure 2): (1) Low, then sharply increasing unemployment (9% of all,) (2) low unemployment (reference, 67%), (3) high unemployment, then slowly decreasing (24%). Thus, both the trajectories 1 and 3 display a clear change in their slopes after the year 1 (after the inclusion year and the first year of follow-up). In more detail, group 1 is characterized by starting with no/low unemployment but having a relatively sharp increase in unemployment and then a stable medium/high level of unemployment until the end of the follow-up. Group 3, in turn, comprised people who had a very high probability of unemployment at the beginning of the follow-up, but whose probability of unemployment gradually decreased then after a couple of years, but was still relatively high at the end of the follow-up. 

In a separate analysis among people with ASD (n = 814), only one unemployment trajectory could be identified (Appendix A). The numbers were small, which partly could explain why several distinct trajectory groups could not be reliably identified. Alternatively, the included young people with ASD followed relatively similar development in their unemployment, and no distinct latent groups could be identified. Thus, none of the models with more than one trajectory group were significant, some of the group sizes were extremely small (e.g., in two groups, 0.5% for one group, and 99.5% for the other). As the further models were not statistically significant, these results were not reliable, and they could have no meaningful interpretation. Hence, ASD was used as a determinant of trajectory membership. 

As displayed in Table 2, people with ASD had higher odds of belonging to the trajectory group 1 (OR 2.53, 95% CI 2.02–3.18) and group 3 (OR 3.60, 95% CI 3.08–4.19). These associations were only slightly reduced after considering all the covariates.

To further examine the determinants of belonging to each of the unemployment trajectories, we assessed which of the sociodemographic and socioeconomic factors were associated with membership of each of the identified trajectory groups. Men were somewhat overrepresented in trajectory group 3 (high then slowly decreasing), while no gender difference was confirmed for the membership in the trajectory group 1 (low then increasing unemployment). Age had an inverse association with trajectory memberships of both groups 1 and 3. Low education, as compared to high education, was particularly strongly associated with a membership in group 3, but also in group 1.

Living in a medium-sized or small town vs. in a large city was also associated with a higher likelihood of belonging to the trajectory groups 1 and 3. Men and those born outside Sweden were somewhat overrepresented in the trajectory group 3.

### 3.3. Mean Unemployment Days

Mean number of unemployment days during the index diagnosis year and the following four years among people with ASD and their references are displayed in Figure 3 as well as in the Appendix A in more detail by diagnosis and trajectory groups. Mean unemployment days were at a higher level among people with ASD as compared to among the reference population at the cohort entry year and during the entire follow-up. In both groups, the levels tended to decrease towards the end of the follow-up, more sharply during the first years. Among people with ASD, the mean number of unemployment days during the year of their diagnosis was 40.1 days (95% CI 35.1–45.1), whereas the corresponding figure in the reference population was 18.4 days (95% CI 17.8–19.0).

### 3.4. Risk of a Disability Pension

The risk of being granted a DP in the years after the five-year unemployment follow-up, was further examined using Cox regression, by the trajectory groups, and by comparing people with ASD to their references from the general population. The overall risk of a DP was very low, 0.43%. However, among people with ASD, it was 9.34% (n=76/814) and in the reference population 0.10% (n=23/22013). Thus, a DP was 93.4 times more likely to be granted during the follow-up among people with ASD who were not on disability pension at baseline or granted DP in the first five years after inclusion (inclusion year included), than in the reference population. 

Trajectory membership was strongly associated with the risk of future DP (Table 3). Thus, belonging to trajectory groups 1 (HR 4.58, 95% CI 2.47; 8.48) or to trajectory group 3 (HR 6.70, 95% CI 4.27; 10.51) was associated with a higher risk of subsequent DP. The associations clearly attenuated after considering ASD (Model 1) for both groups 1 (attenuation 45.8%) and 3 (attenuation 46.5%). The associations further attenuated somewhat after considering birth country, type of living area, and educational level (Model 2, attenuation 16.1% and 9.1%, respectively). However, all the associations remained statistically significant.

## 4. Discussion

### 4.1. Main Findings

This exploratory study sought to identify future unemployment trajectories among young adults. We identified three distinct such trajectories of unemployment, describing different developmental patterns of unemployment among young people: (1) stable low probability of unemployment, (2) low then sharply increasing probability, and (3) high and slowly decreasing probability of unemployment. Having an ASD diagnosis was strongly associated with a higher probability of being in a high-unemployment trajectory, but it is of note that the number of mean days of unemployment per year was not high in any of the trajectories. Finally, also the absolute risk of a DP after the unemployment follow-up was very low in this cohort. However, memberships in a high-unemployment trajectory strongly predicted the relative risk of a subsequent disability pension. The risk of a disability pension was also notably higher among people with ASD, as compared to among their matched references from the general population.

### 4.2. Interpretation

There has been little previous research on unemployment and disability pension risk among people with high functioning ASD, who are not receiving a disability pension. Particularly, person-oriented methods are seldom used. Mostly, the focus has rather been on a higher risk of adverse health outcomes among people with ASD, such as of premature death [12] and even suicidality [23]. The risk of premature death concerns particularly the “low functioning” ASD people with an intellectual disability, but to some extent also “high functioning” ASD cases. These outcomes, although not directly measuring employment, still represent part of the reasons for not being employed, and have been previously coded e.g., as “unsuccessfully employed” among people with ASD [24]. Our parallel study also highlights the very high risk of sickness absence and disability pension among high functioning ASD people, as compared to their matched references from the general population [25]. However, these previous studies did not specifically focus on ASD people who were not on a disability pension, and therefore, their developmental patterns of unemployment and subsequent risk of a disability pension, in comparison to the general population, is not known. As ASD people were a small part (3.6%) of the cohort, we show factors that are associated with disability among young adults in general although we also studied the developmental patterns in a separate analysis and used ASD as a predictor in the models.

While ASD can be a highly disabling condition for some people, it is important to highlight that there is great variation and wide spectrum of “symptoms”. Thus, it is important to shed light on those people with ASD who are in paid work or equivalent (e.g., students), to help support their participation in paid work, to prevent increasing unemployment and the risk of later disability pension.

A German study showed that adults with a late-diagnosed ASD (mean age at diagnosis 35 years) have more unemployment and a higher likelihood of early exit from paid employment due to health problems [7]. Although that study was cross-sectional and had a low response rate (43%; n = 185), the results were similar to our register-based study among people with early ASD onset. Both studies thus highlight the overall disadvantaged situation of people with ASD in terms of their employment patters as compared to the general population. A review comprising different outcomes of ASD showed that people with ASD have limited social integration, poor job prospects and more mental disorders, although the included studies were very heterogeneous e.g., regarding their methodology [10]. In all, it is important to consider these broader social and health-related characteristics, as factors such as inadequate social skills or difficulties regarding independence may contribute to unstable careers, as indicated e.g., by high unemployment rates. Accordingly, the National Institute for Health and Clinical Excellence (NICE) guidelines in UK on ASD recognition, referral, diagnosis and management also highlight that particularly broad support measures are needed and that the care of people with ASD should not only focus on mental health [26,27]. As the assessment criteria of possible ASD include difficulty in obtaining and sustaining employment (or education), the guidelines further suggest supported employment programmes to thoroughly focus on all aspects of successful job integration, including help in the job application process, and matching the people with ASD with the job, adjusting working conditions to their needs, specific training in jobs, etc. [28].

It would have been interesting to observe distinct unemployment trajectories separately among people with ASD, and if possible, to identify any groups among people with ASD that have sustained low unemployment, reflecting stable employment. However, many people with ASD are not in paid work, and those who are, appear to share a higher risk of receiving a DP during young adulthood. Identifying just one trajectory, if approximating the true situation, is also a noteworthy result, suggesting that there are no distinct trajectories among employed young people with ASD, but those who work are likely to be a relatively homogenous group in terms of their developmental patterns in unemployment. Thus, this sheds light and helps better understand (un)employment and disability pension risks among people with ASD.

While this and some previous studies highlight an overall higher likelihood of unemployment and the risk of disability pension among people with ASD as compared to their matched references, a study from the United Kingdom showed that supported employment is a cost-effective intervention for them [29]. Although it can be initially costly, in the long run such support could promote the well-being of people with ASD and therefore also equal opportunities in society, and reduce cost of unemployment, and particularly early exit from paid employment through disability pension. The need for supported employment to better and more adequately meet the specific needs of people with ASD and their degree of disability has been highlighted [11]. Since ASD often is diagnosed early in life, life course approaches are needed, as well as early and specifically targeted interventions. More studies should address their efficacy in the long run, e.g., in supporting stable employment and health and well-being of people with ASD.

### 4.3. Methodological Considerations

A strength of this study is the use of linked nationwide, population-based register data of high quality [15,30,31,32,33]. We were thus able to include a comprehensive cohort of all people in Sweden diagnosed with ASD during the study period. If a person had been diagnosed elsewhere or out of Sweden, we would miss them, but as our inclusion criteria stated that all had to have lived in Sweden at least five years before their diagnosis or cohort entry, it is unlikely to be a major issue. Another strength is the inclusion of matched references from the whole population, who did not have any record of mental disorders (F diagnoses) during the entire follow-up (including ASD). The matching was done so that five matched references were randomly selected for each individual with ASD. Furthermore, after matching, there should be no differences in the original full cohort by the studied key social factors. However, it is of note that as we had several inclusion and exclusion criteria in this study, there are some differences, as shown in the results. As we focused on unemployment, many people with ASD were excluded, because they either were on a DP at inclusion, or exited paid work during the unemployment follow-up where five years with no disability pension nor death was required (inclusion year and four years after). In contrast, the proportion of the matched references initially on a DP was extremely low, thus excluding those on a DP has little effect on the distributions of social factors among the references. When only including those people with ASD who were at risk of unemployment during their diagnosis year and the following four years, it is important to understand their profile, as their social characteristics are no longer the same as compared to all people with ASD.

The absolute number of unemployment days annually, and the risk of receiving a DP during follow-up was in general low, possibly due to the young age of the included people with and without ASD (up to 35 years). As the trajectories were made among the population with five referents and the cases, the referents’ unemployment days dominated in each trajectory. The trajectories approximate probability of unemployment annually, and mean days were computed to describe the differences between the trajectory groups. Due to the design and the focus on young adults, the overall risk of DP was low. The low risk of DP partly reflects a short follow-up time in terms of the entire working life span, and the inclusion of only young adults. The relative risk of DP among people with ASD was, however, high, as compared to their references. Still, in all, we showed that there is a large proportion of people with ASD, who are *not* on DP, and even if they belonged to the trajectory of high unemployment, the mean annual unemployment days were not very high, and the absolute risk of a DP after the unemployment follow-up was rather low. More detailed elaboration of the actual reasons behind sustained work was, however, beyond the scope of this study.

Finally, some further limitations have to be acknowledged. It is a limitation of this register-based study that we do not know about the type and quality of work or job type, e.g., there is no information about hours worked, salary levels, and types of employment contract, psychosocial and physical work characteristics, and whether these differed between people with ASD and their references. A previous large study showed that people with ASD worked notably fewer hours, and also earned less [24]. Such factors could partly contribute to the differences in the probability of unemployment, but we could not take them into account in this study. We further cannot distinguish between any employment types such as supported employment, sheltered employment, self- and home-based employment or other types of the employment, which could also be considered as a paid employment. However, as our focus was rather on the developmental patters in unemployment, using the information on benefits in the available register data, these different types of employment are out of the scope of the current study. In other words, we can only assess unemployment, if there are such registered benefits, and our focus is not on employment per se. Additionally, it needs to be emphasized that, as the trajectories are but approximations of the true development, we cannot rule out that some people were misclassified and the group where they were placed does not describe the true development of their unemployment. However, as the reliabilities in each trajectory group were acceptable or high (mean posterior probability was 0.79 for trajectory group 1, 0.92 for trajectory group 2, and 0.96 for trajectory group 3), a classification error is unlikely to affect our results to any large extent. Finally, a limitation or a point to note is that the results do not generalize to all young people with ASD, due to the above-mentioned applied exclusion and inclusion criteria. However, the study was specifically set to focus on a population who can be working, which is also a value of the study. With this focus, we provide new information about unemployment and disability pension risks of young adults with ASD, and their references.

## 5. Conclusions

In our study of unemployment trajectories and the future risk of DPs among young adults, the predictions are largely driven by ASD independent factors, because ASD was relatively rare in the analysed cohort. However, young adults diagnosed with ASD, who at inclusion were not on a disability pension (DP), had a higher probability of belonging to a future trajectory with higher unemployment, and later a higher risk of a disability pension than young adults without mental disorders. To help people with disabling conditions such as ASD to enter paid work, and to have more stable work careers, more studies are needed, such as those that could focus e.g., on early intervention and the role of comorbid conditions. Moreover, groups such as policymakers, employers and other stakeholders could aim to address these results. Given that the severity and symptoms of ASDs vary significantly between persons [10], any single common measure is unlikely to be effective in preventing unemployment and disability pension. As the results highlight that most of the people with ASD in this study had a relatively low mean number of unemployment days, and were not on a disability pension at inclusion nor during the first five follow-up years, more targeted studies are needed to gain knowledge on what promotes and facilitates their participation in paid employment in young adulthood and later during the working life span.

## Figures and Tables

**Figure 1 ijerph-17-02486-f001:**
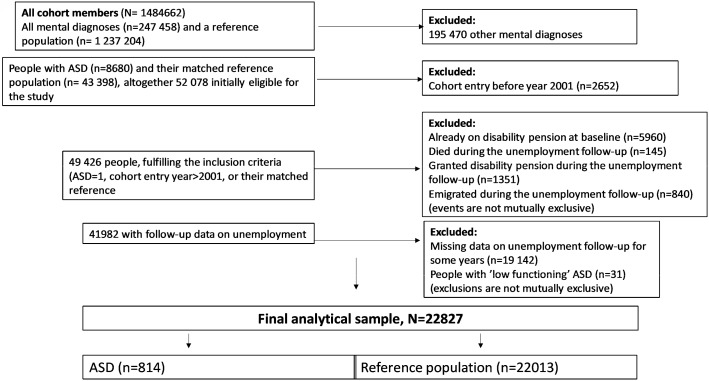
Formation of the data, reasons for exclusion, and n of participants (in each data phase).

**Figure 2 ijerph-17-02486-f002:**
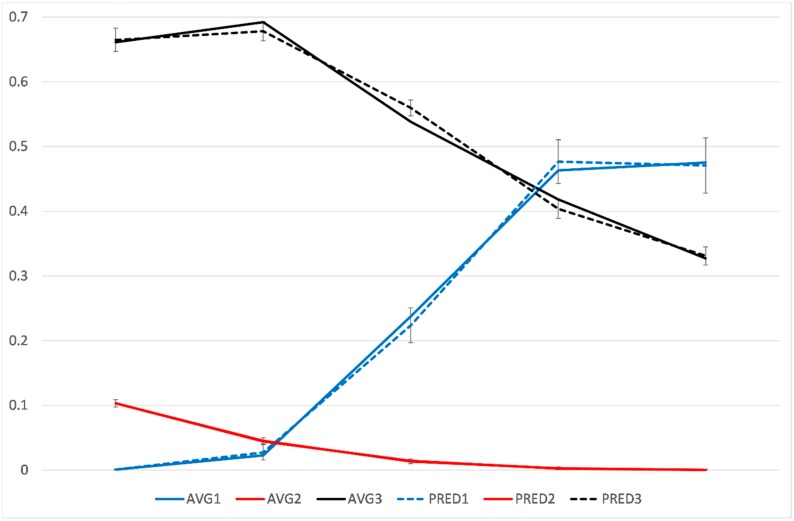
Unemployment trajectories during the follow-up (cohort entry year, 0, and a four-year follow-up) among individuals in the study population (autism spectrum disorder, ASD, 3.6%). AVG = average, PRED = predicted for each trajectory group. PRED1, PRED2, AND PRED3 (dashed lines) are the model estimates with their 95% confidence intervals for trajectory groups 1, 2, and 3, respectively. AVG2 and PRED2 overlap almost completely. *Y* axis: probability of unemployment; *X*-axis: follow-up years from entry.

**Figure 3 ijerph-17-02486-f003:**
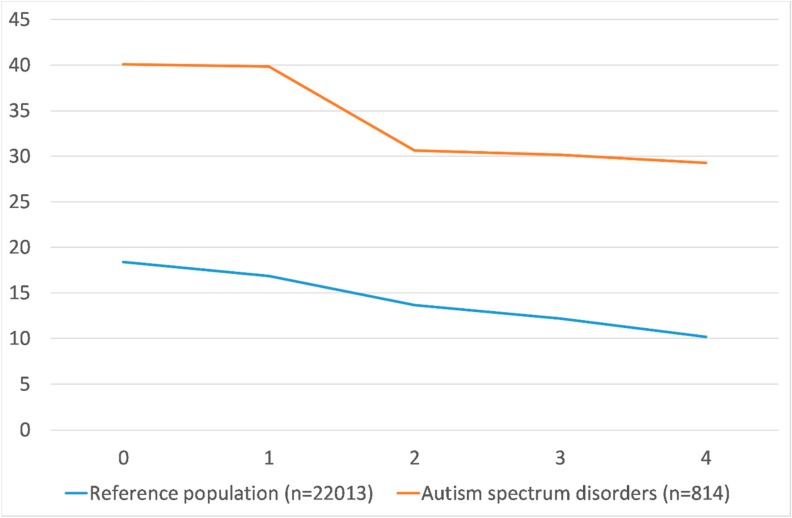
Mean unemployment days (*y*-axis) during the cohort entry year and four years after (*x*-axis), among people with autism spectrum disorder (ASD) and in their matched references from the general population.

**Table 1 ijerph-17-02486-t001:** Descriptive characteristics of cohort members, and by people with autism spectrum disorder (ASD) and in the reference population.

Sociodemographics and Trajectory Groups	All (n = 22,827)		Reference Population (n = 22,013), %		ASD (n = 814), %		*p*-Value (Difference between ASD Individuals and References)
	Frequency	Percent	Frequency	Percent	Frequency	Percent	
**Sex**							
Men	16049	70.3	15452	70.2	597	73.3	
Women	6778	29.7	6561	29.8	217	26.7	0.0537
**Educational level**							
Low	3520	15.4	3247	14.8	273	33.5	
Intermediate	15363	67.3	14947	67.9	416	51.1	
High	3944	17.3	3819	17.4	125	15.4	<0.0001
**Family situation**							
Married or cohabiting	2642	11.6	2605	11.8	37	4.6	
Single or living at home with parents (youth)	20185	88.4	19408	88.2	777	95.5	<0.0001
**The type of living area at follow-up start**							
Big cities	8204	35.9	7917	36.0	287	35.3	
Medium-sized cities	7712	33.8	7431	33.8	281	34.5	
Rural areas	6911	30.3	6665	30.3	246	30.2	0.8845
**Birth country**							
Born in Sweden	21106	92.5	20,343	92.4	763	93.7	
Born outside Sweden	1721	7.5	1670	7.6	51	6.3	0.161
**Trajectory group**							
Low, then sharply increasing unemployment	1883	8.3	1782	8.1	101	12.4	
Low unemployment (reference)	16,247	71.2	15,881	72.1	366	45.0	
High unemployment, then slowly decreasing	4697	20.6	4350	19.8	347	42.6	<0.0001

**Table 2 ijerph-17-02486-t002:** Autism spectrum disorder (ASD) and social determinants of trajectory membership ((1) Low, then sharply increasing unemployment (2) low unemployment (reference), (3) high unemployment, then slowly decreasing): odds ratios (OR) and their 95% confidence intervals (95% CI) from multinomial logistic regression analyses.

Autism Spectrum Disorder (ASD) and Sociodemographics		Age- and Sex-Adjusted Models			Fully Adjusted Model		
	Trajectory Group	Odds Ratio	95% CI		Odds Ratio	95% CI	
**Autism spectrum disorder (ASD) as a determinant of trajectory memberships**	**Trajectory group**						
Reference	Low unemployment	1			1		
ASD vs. reference population	High then slowly decreasing	3.60	3.08	4.19	3.23	2.75	3.79
ASD vs. reference population	Low then sharply increasing	2.53	2.02	3.18	2.42	1.92	3.04
**Social determinants’ associations with trajectory memberships**							
Reference	Low unemployment	1			1		
Men vs. women	High then slowly decreasing	1.28	1.19	1.38	1.24	1.15	1.34
Men vs. women	Low then sharply increasing	0.98	0.88	1.08	0.97	0.87	1.08
Reference	Low unemployment	1			1		
Age (continuous)	High then slowly decreasing	0.91	0.91	0.92	0.94	0.93	0.95
Age (continuous)	Low then sharply increasing	0.95	0.94	0.96	0.95	0.94	0.96
Reference	Low unemployment	1			1		
Low education vs. high education	High then slowly decreasing	4.92	4.28	5.66	4.25	3.69	4.90
Low education vs. high education	Low then sharply increasing	1.99	1.66	2.39	1.84	1.53	2.21
Intermediate education vs. high education	High then slowly decreasing	1.86	1.64	2.11	1.69	1.49	1.92
Intermediate education vs. high education	Low then sharply increasing	1.10	0.95	1.28	1.07	0.92	1.25
Reference	Low unemployment						
Medium-sized town vs Large city	High then slowly decreasing	1.68	1.55	1.83	1.76	1.62	1.92
Medium-sized town vs Large city	Low then sharply increasing	1.37	1.22	1.53	1.41	1.25	1.58
Small town/village vs Large city	High then slowly decreasing	2.06	1.90	2.24	2.14	1.96	2.33
Small town/village vs Large city	Low then sharply increasing	1.25	1.11	1.41	1.30	1.15	1.46
Reference	Low unemployment	1			1		
Born outside Sweden vs. in Sweden	High then slowly decreasing	1.86	1.73	2.00	1.80	1.60	2.03
Born outside Sweden vs. in Sweden	Low then sharply increasing	1.31	1.18	1.45	1.71	1.45	2.02

**Table 3 ijerph-17-02486-t003:** Trajectory group membership ((1) Low, then sharply increasing unemployment (2) low unemployment (reference), (3) high unemployment, then slowly decreasing) as a predictor of subsequent risk of a disability pension. Hazard ratios, HR, and their 95% confidence intervals, 95% CI.

	Crude Model (Model 0)			Age- and ASD-Adjusted Model (Model 1)			Fully Adjusted Model (Model 2)		
Total population	HR	95% CI		HR	95% CI		HR	95% CI	
Trajectory group									
Low unemployment (reference)	1.00			1.00			1.00		
High then slowly decreasing	6.70	4.27	10.51	2.77	1.75	4.39	2.52	1.58	4.02
Low then sharply increasing	4.58	2.47	8.48	2.28	1.23	4.24	2.00	1.07	3.74
**Autism spectrum disorder (own effect)**									
No				1.00			1.00		
Yes				92.72	57.12	150.51	83.71	51.23	136.77

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
