# Peer review of "Unemployment Trajectories and the Early Risk of Disability Pension among Young People with and without Autism Spectrum Disorder: A Nationwide Study in Sweden"

_ijerph, 2020, doi:10.3390/ijerph17072486_

Round 1
Reviewer 1 Report
This is an updated and more informative version of the previously submitted manuscript. The authors have done all their best to address different issues that were raised in the first submission.
Reviewer 2 Report
Lallukka and coauthors have addressed all comments to a satisfactorily degree.
Reviewer 3 Report
Thank you for asking me to review the revisions made to your manuscript. I reviewed the revisions, and the authors addressed all of my comments in a clear and concise manner. As a result, the manuscript is much stronger now. Thank you for your work in this important area, and I wish you all the best going forward.
This manuscript is a resubmission of an earlier submission. The following is a list of the peer review reports and author responses from that submission.
Round 1
Reviewer 1 Report
This is a very interesting study on the issue of identifying unemployment trajectories among young adults with ASD in a developed country with a high level of social supports. I think this article has the potential to be a valuable contribution to the Journal. The focus on unemployment among people with ASD is of particular relevance in affluent countries such as Sweden or other northern European countries. Hence, as a paper for potential merit of being published in an international journal some more clarification is needed. This is very odd that the risk of unemployment and being granted disability pension considered equally. I think this is a very important issue and needs some clarification. The paper in its present form would have less relevance for an international readership although the substance would be relevant.
Generally, I recommend explanation and definitions of terms such as “disability pension” and its associated challenges (since instead of “possibilities” of granting a DP it is mentioned as a “risk”) for the international readers. Disability pension has internationally been considered as social support and deserved to be explained in more details about its possible associated challenges.
Employment also need to be more explained there are different types of employment. It is not clear if supported employment, sheltered employment, self and home-based employment and other types of the employment which all has the potential to be considered as a paid work, has been considered in this study or a special definition of employment.
The other important issue from my point of view is the problem with readership and implication of the findings. The authors might be able to identify the group they wish to address. Are they policymakers, employers or other stakeholders?
Another possible contributing factor with determining the readers group might be the considered theoretical framework. This is also find to understand which theoretical framework has been followed in this study. In the interpretation part it is indicated that a person-oriented method has been considered for the study but more data regarding this approach and the paradigm which this approach has been rooted form is needed.
In sum, the paper is well written and might be restructured into a more applicable one in the light of the above mentioned points and the following issues I encountered during the review process.
Some minor points:
I recommend the ICD-10 diagnosis codes to be explained.
In the result part lines 179 and 180 reporting SD will help to distribution of the sample and the amount of variation or dispersion of the sample.
Line 182 is difficult to follow, I recommend a revision (is the word “reference” addressing the members of the matched group?):
Of the people with ASD, less than 5% were married, and about 10% of their references.
Line 276 indicated that “Although ASD is a highly disabling condition” this is a very strong judgment and needs some more justification and references because it covers the entire group under the umbrella of ASDs.
Author Response
Response to Reviewer 1 Comments
Comments and Suggestions for Authors
Point 1: This is a very interesting study on the issue of identifying unemployment trajectories among young adults with ASD in a developed country with a high level of social supports. I think this article has the potential to be a valuable contribution to the Journal. The focus on unemployment among people with ASD is of particular relevance in affluent countries such as Sweden or other northern European countries. Hence, as a paper for potential merit of being published in an international journal some more clarification is needed. This is very odd that the risk of unemployment and being granted disability pension considered equally. I think this is a very important issue and needs some clarification. The paper in its present form would have less relevance for an international readership although the substance would be relevant.
Response 1: Thank you for your positive comments. From your comments we realised that we had not been clear enough in the description of the study design and have, therefore, revised the text throughout about this.
Our focus was on unemployment among young adults diagnosed with ASD, and their future risk of disability pension after the unemployment follow-up. Therefore, we only included those who were not on disability pension when included, nor during the five-year unemployment follow-up. Thus, the disability pension follow-up started after the five-year unemployment follow-up had ended.
Indeed, we are not considering unemployment and disability pension as equal, but they are included from different points in time. The Swedish insurance system further does not allow simultaneous unemployment and work disability. We hope that the paper is now clearer with this respect, and better highlights the rationale to include both unemployment and disability pension in this paper.
Please see abstract and main text for clarifications to highlight the assumed relationships between unemployment and disability pension in this study (throughout the manuscript, e.g. abstract aims, lines 19-21; introduction lines 78-91, Methods lines 151-164)
Point 2: Generally, I recommend explanation and definitions of terms such as “disability pension” and its associated challenges (since instead of “possibilities” of granting a DP it is mentioned as a “risk”) for the international readers. Disability pension has internationally been considered as social support and deserved to be explained in more details about its possible associated challenges.
Response 2: Thank you for alerting us on having missed to provide such information. This has now been done in the revised manuscript. Disability pension can be granted people aged 19-64 years who live in Sweden and have permanent or long-term reduced work capacity due to injury or disease. The DP benefit can be granted for full-time or part-time of ordinary work hours, and amounts to 64% of lost income, up to a certain level. People with no previous income get a lower benefit. People actively seeking job can get unemployment benefit even if not having had previous income (please see lines 110-114).
Point 3: Employment also need to be more explained there are different types of employment. It is not clear if supported employment, sheltered employment, self and home-based employment and other types of the employment which all has the potential to be considered as a paid work, has been considered in this study or a special definition of employment.
Response 3: Thank you for alerting us on not having been precise about this. We agree that it is important to clarify what employment means in this study. Our focus is actually on unemployment benefits and on disability pension. More importantly, we do not study employment directly, but we identified unemployment trajectories and used them as predictors of disability pension, indicating permanent exit from paid employment. We have now checked and revised the manuscript throughout in this respect, to be clearer that we focus on first identifying unemployment trajectories, and second use them as a determinant of subsequent disability pension risk, indicating exit from the possibility of paid employment. In the study we used register data regarding information on e.g. unemployment days and disability pension days. We did not have information on any type of paid work, nor about being active in other ways, e.g., through being a student, on parental leave etc, as that was not the focus of this study. This was not in the scope of this study, and warrants other type of research. We have clarified these points in the revised discussion (please see lines 416-427, and all tracked changes referring to unemployment and disability pension). Please also refer to our Response #1.
Point 4: The other important issue from my point of view is the problem with readership and implication of the findings. The authors might be able to identify the group they wish to address. Are they policymakers, employers or other stakeholders?
Response 4: We appreciate the comment. This is a first, exploratory study regarding future unemployment, and later, disability pension, among people with ASD compared to population-based references – in order to get a better insight into this. So far, the results are very general. Hopefully they can inspire future studies, leading to information for different stakeholders. We have also added the suggested groups that could potentially address our results (please see revised Conclusions). However, definite implications cannot be based on results from this type of explorative study – they would be speculations, which are not appreciated in a scientific journal. Thus, we prefer to be cautious and avoid extrapolation and conclusions that are not supported by the data.
Point 5: Another possible contributing factor with determining the readers group might be the considered theoretical framework. This is also find to understand which theoretical framework has been followed in this study. In the interpretation part it is indicated that a person-oriented method has been considered for the study but more data regarding this approach and the paradigm which this approach has been rooted form is needed.
In sum, the paper is well written and might be restructured into a more applicable one in the light of the above mentioned points and the following issues I encountered during the review process.
Response 5: We have added more details about person-oriented methods, namely group-based trajectory analyses (GBTA) and the rationale and added value to use them in this study. It is a statistical method to address our aims, and an alternative to more widely and traditionally used variable-oriented approaches to examine e.g. the risk of disability pension. Thus, our study is based on epidemiological theory, where we use baseline information to study future situations. Otherwise, this is not a theory-driven study, but rather a data-driven, explorative register-based epidemiological study. Our introduction aims to show and briefly sum what is known about this area and what this study aims to add, using the person-oriented method. GBTA can suggest a certain number of discrete latent trajectories in observational data, such as ours, however, it cannot be concluded that such discrete trajectories exist. Instead, they are approximations of true development and the usefulness of the classification can bet assessed e.g. through validation and replication studies. We have amended the introduction discussion to add more (methodological) consideration regarding the GBTA (please see lines 70-75 and 418-427).
Some minor points:
Point 6: I recommend the ICD-10 diagnosis codes to be explained.
Response 6: We have now added explanations for the ICD-10 codes in the revised paper for ASD: F84.0, Childhood autism; F84.1. Atypical autism; F84.3. Other childhood disintegrative disorder; F84.5. Asperger's syndrome; F84.8. Other pervasive developmental disorders; F84.9. Pervasive developmental disorder, unspecified (please see lines 119-124).
For mental retardation, F70-70 the explanations are as follows: F70 Mild mental retardation; F71 Moderate mental retardation; F72 Severe mental retardation; F73 Profound mental retardation; F78 Other mental retardation; F79 Unspecified mental retardation (please see lines 142-144).
To be included as a reference, the individuals were also required to have been living in Sweden for at least five years before their cohort entry, and to have no indication of any mental disorders (ICD-10 codes F00-F99) in the available patient registers throughout the entire follow-up period (from 1987 to 2013). We think it is too much to open up all F codes, but have opened up all those relevant for ASD and the exclusion criteria (mental retardation).
Point 7: In the result part lines 179 and 180 reporting SD will help to distribution of the sample and the amount of variation or dispersion of the sample.
Response 7: We have now added the SDs for mean age, as suggested (lines 206-207).
Point 8: Line 182 is difficult to follow, I recommend a revision (is the word “reference” addressing the members of the matched group?):
Response 8: Yes, the word references here means matched references from the general population. We tried to avoid repeating the very long version too many times. For clarity, we have added this here in the beginning of the result section.
Point 9: Of the people with ASD, less than 5% were married, and about 10% of their references.
Response 9: This is correct, and references also here mean matched references from the general population. We hope that the revised version is clearer, as this is mentioned couple of lines before. To avoid repetition, this sentence is retained. The numbers, of course, indicate something about the possibility or the urge to be in a married couple and would be interesting to study further in another paper.
Point 10: Line 276 indicated that “Although ASD is a highly disabling condition” this is a very strong judgment and needs some more justification and references because it covers the entire group under the umbrella of ASDs.
Response 10: We agree and have now clarified and toned down the sentence, as suggested. It is true that people with ASD have very varying levels of functioning, with some needing much help and others being independent, having mainly specific social problems. In our study, we have a population who do not represent or are intended to represent, all people with ASD, as people with mental retardation and people who were on full disability pension at the cohort entry, were excluded. This is because their unemployment days are zero throughout the follow-up, but for a completely different reasons as compared to e.g. those who are all the time in paid work. As ASD is often seen as a disabling condition, our study shows that many people with ASD have relatively low number of days unemployment per year, and a relatively low absolute risk of disability pension during young adulthood. Nonetheless, disability pension event after unemployment follow-up among people with ASD still is almost a hundred times more likely as compared to among people without ASD, even in a population that at cohort entry includes only people who were not on disability pension at the cohort entry nor during the entire unemployment follow-up (cohort entry year plus 4 years after that).

Reviewer 2 Report
The authors of the current manuscript deal with the interesting question on how integrated high functioning ASD individuals are on the job market by assessing the level of unemployment and disability pension rate. A sample of 814 ASD individuals are related to roughly 22,000 matched controls having no mental disorder. Trajectory differences based on unemployment probabilities as a function of time where found along with an increased risk of receiving disability pension among ASD individuals within the time period studied. The manuscript is well structured and written, I have the following comments:
- ASD also belongs to the umbrella category “mental disorders”, please revise line 43-45
- The authors should include a time-line giving the reader a good overview of e.g. time of diagnosis, ascertainment period, follow up period (section 2.1.)
- Labels missing in figures
- Could the authors elaborate on the fact that trajectory 1 and 3 display a dramatic change in slope after year 1.
- There seems to be no 95% CI included in the graph.
- Line 206; the trajectory should be shown. How does this compare to trajectory 1 and 3? If diverging substantially, then the predictions are driven by ASD independent factors and hence not very useful.
- Although a lack of descriptive variables is limiting the interpretations, is there a difference between ASD (or reference) individuals belonging to trajectory 1 or 3? Thoughts should be included in the manuscript.
- Can the authors account their analyses for job type?
- The authors should elaborate more on line 208-209. E.g. including tables or graphs.
- How do the authors interpret the association between low education level and trajectory 1 and 3, considering that the trajectories are anti-correlated?
- Authors should show Cox regression hazard graphs (are assumptions fulfilled?)
- Minor text editing is required e.g. line 352 “to aim to understand” and line 372 “thus”
Author Response
Response to Reviewer 2 Comments
Comments and Suggestions for Authors
The authors of the current manuscript deal with the interesting question on how integrated high functioning ASD individuals are on the job market by assessing the level of unemployment and disability pension rate. A sample of 814 ASD individuals are related to roughly 22,000 matched controls having no mental disorder. Trajectory differences based on unemployment probabilities as a function of time where found along with an increased risk of receiving disability pension among ASD individuals within the time period studied. The manuscript is well structured and written, I have the following comments:
Response: Thank you for these positive comments.
Point 1: ASD also belongs to the umbrella category “mental disorders”, please revise line 43-45
Response 1: We have now revised these lines to be clearer with this respect, as suggested. The idea was to highlight that depression and anxiety as ‘common mental disorders’ have been in the focus of research more often than other mental disorders, including ASD (please see lines 45-48).
Point 2: The authors should include a time-line giving the reader a good overview of e.g. time of diagnosis, ascertainment period, follow up period (section 2.1.)
Response 2: We have now clarified the methods section, as suggested (please see lines 118-145, tracked parts in particular). ASD was indicated based on the first date of recorded ICD-10 codes available in in- and specialised outpatient health care during 2001–2010. As ASD is very rarely missed, we are likely to have included all individuals diagnosed with ASD during the study period, provided the inclusion criteria were filled in (Figure 1).
Point 3: Labels missing in figures
Response 3: We have checked that all the figures have labels in the revised manuscript. The figure 3 was revised also otherwise (e.g. to include 95% confidence intervals, as suggested), and there is a new appendix figure, which previously was reported as ‘data not shown’.
Point 4: Could the authors elaborate on the fact that trajectory 1 and 3 display a dramatic change in slope after year 1.
Response 4: We agree that those changes are dramatic. Possible reasons could be related to some unmeasured individual level characteristics, for example. If we had e.g. survey data or more self-reported detailed information available about other changes that might happen during or after year 1, this would help in such an elaboration. Unfortunately, further elaboration regarding reasons for such a change, is not possible in these data or within this study, and it would be speculative. We have, however, added the reviewer’s point about the changes in the slopes after year 1 in the revised results (please see lines 219-225).
Point 5: There seems to be no 95% CI included in the graph.
Response 5: You are right. We have now included 95% CI in our Figure 3. They are not very wide, and almost invisible for the largest trajectory group 2. We also now used same colours for average and estimated (predicted) figures for clarity. Thus, AVG1, AVG2 and AVG3 are the averages and PRED1, PRED2 AND PRED3 are the model estimates with their 95% CIs. Averages are shown with a solid line, and model estimates are shown using a dashed line. For the reference group (trajectory 2), the lines overlap almost entirely.
Point 6: Line 206; the trajectory should be shown. How does this compare to trajectory 1 and 3? If diverging substantially, then the predictions are driven by ASD independent factors and hence not very useful.
Response 6: We have now clarified the results regarding the models examining only people with ASD (please see lines 232-239). In a separate analysis among people with ASD only (n=814), there was only one trajectory that was statistically significant. It cannot be compared to the trajectories shown for current models including the matched references, as the study populations are completely different. Thus, only those trajectories can be compared that are from the same analysis and sample. We hope that the revised version avoids confusion regarding this aspect. The trajectory among people with ASD has now been added as an Appendix. None of the models with more than one group could be used (none of them was significant and they cannot be displayed, interpreted, or used for the analyses). Based on our analysis, it is likely that most people with ASD follow a similar trajectory in their developmental patterns of unemployment over time.
Additionally, we have added as a limitations and revised our conclusions to be clearer that as ASD individuals are a small part of the study population, the predictions are driven by ASD independent factors (lines 317-319 and revised conclusions, first sentence).
Point 7: Although a lack of descriptive variables is limiting the interpretations, is there a difference between ASD (or reference) individuals belonging to trajectory 1 or 3? Thoughts should be included in the manuscript.
Response 7: This can be first seen in the last rows of Table 1. P-value is significant which implies that people with ASD indeed are more likely to belong to the trajectory groups 1 and 3, as compared to their matched references. Second, Table 2 confirms that both after adjusting for sex and age as well after full adjustment (first rows of the Table 2), individuals with ASD are more likely to belong to trajectory 1 and 3. Thus, odds of belong to the trajectory 1 and 3 ranged from 2.4 to 3.2 in the fully adjusted models, for trajectories 1 and 3, respectively.
The confidence intervals somewhat overlap so it cannot be proven if the association of ASD is stronger for trajectory 1 or 3. These results are included in the manuscript, just before the Table 2.
Point 8: Can the authors account their analyses for job type?
Response 8: Unfortunately, we do not have data for job type nor any other information about paid work or other activities, apart from unemployment benefits and DP. We have acknowledged this in the discussion (please see line 411).
Point 9: The authors should elaborate more on line 208-209. E.g. including tables or graphs.
Response 9: We have now opened up the results in more detail. We have also added a new Supplementary figure (please see lines 232-239 and a new Figure S1).
Point 10: How do the authors interpret the association between low education level and trajectory 1 and 3, considering that the trajectories are anti-correlated?
Response 10: Both trajectories 1 and 3 comprise more people with unemployment days, as compared to trajectory 2, which is characterized with very low or no unemployment. Mean number of unemployment days in each trajectory are in the supplementary table S2. The table further shows that the level of unemployment is notably higher in the trajectory group 3 than 1. As low education is related to unemployment, it is understandable and expected that it is associated with both the membership to trajectory 1 and 3. Furthermore, it is also expected that the associations are stronger for trajectory 3, considering that the mean number of unemployment days is notably higher in this group than in the group 1.
Point 11: Authors should show Cox regression hazard graphs (are assumptions fulfilled?)
Response 11: Yes, the assumptions are fulfilled. This is reported in the Statistical analyses section, please see lines 200-202:
We also tested whether there was any indication for Cox proportional hazards violation. The proportionality test suggested that there was no violation against the assumptions (p-value for the overall test checking the proportionality assumption was 0.227).
Point 12: Minor text editing is required e.g. line 352 “to aim to understand” and line 372 “thus”
Response 12: These have been corrected, thank you. We have also scrutinized the text for other language aspects and revised accordingly.

Reviewer 3 Report
Thank you for allowing me to review your manuscript titled “Unemployment trajectories and the early risk of disability pension among young people with and without autism spectrum disorders: a nationwide study in Sweden.” The following are my comments, section by section, that I believe will strengthen the manuscript.
Title:
- The title is appropriate; however, I suggest changing “disorders” to “disorder” to be consistent with ICD-10 language. I also recommend making this change throughout the manuscript.
Abstract:
- The abstract is clear and concise. I have no comments.
Introduction: The Introduction is well-written and clear.
- Line 49 - suggest changing “challengers” to “challenges.”
Materials and Methods:
- Do you have information on how ASD diagnoses were made? If so it may be useful. If not, it is not a large issue. Given the variability in how ASD is diagnosed, it is prudent to include this information if available.
- Was race considered as a covariate?
Results:
- What was the proportion of men/women with ASD (line 186)?
- For Figure 2, it would be useful to label the 3 distinctive trajectories rather than Avg 1, Avg2, etc. in the figure caption just to be clear. Also, please label x and y axes
- For Figure 3, please label x and y axes in the figure so it is clear at a glance.
Discussion:
- The discussion is appropriate and informational.
- My only suggestion would be to consider the impact of co-occurring conditions on unemployment in those with ASD. Anxiety and depression are mentioned in the Introduction but not mentioned again. Many individuals with ASD have co-occurring anxiety disorders which can limit suitability for work. Further, physical symptoms like gastrointestinal disorders can limit one’s ability to work, and they are quite common in ASD. Perhaps these could be considered for future studies - to highlight the importance of treating co-occurring conditions if they impact employment in those with ASD.
Conclusions
- The conclutions are appropriate. I have no further comment.
Overall, this is a very well written manuscript and equally well conducted study. I do not have many comments. I congratulate the authors on their important work and wish them well with their future studies to help those with ASD.
Author Response
Response to Reviewer 3 Comments
Comments and Suggestions for Authors
Thank you for allowing me to review your manuscript titled “Unemployment trajectories and the early risk of disability pension among young people with and without autism spectrum disorders: a nationwide study in Sweden.” The following are my comments, section by section, that I believe will strengthen the manuscript.
Response: Thank you for these positive comments.
Point 1: Title:
- The title is appropriate; however, I suggest changing “disorders” to “disorder” to be consistent with ICD-10 language. I also recommend making this change throughout the manuscript.
Response 1: We have made the change, as suggested. All autism spectrum disorders have been changed to read autism spectrum disorder.
Point 2: Abstract:
- The abstract is clear and concise. I have no comments.
Response 2: Thank you.
Point 3: Introduction: The Introduction is well-written and clear.
- Line 49 - suggest changing “challengers” to “challenges.”
Response 3: Thank you. We have corrected the mistake.
Point 4: Materials and Methods:
- Do you have information on how ASD diagnoses were made? If so it may be useful. If not, it is not a large issue. Given the variability in how ASD is diagnosed, it is prudent to include this information if available.
- Was race considered as a covariate?
Response 4: We have clarified the description of ASD diagnoses, and added all the ICD-10 codes used (please see lines 118-124).
Regarding, race, we only have information about the country of birth, which was used in matching. In the analyses, we only used a dichotomous variable (born in Sweden or outside Sweden).
Point 5: Results:
- What was the proportion of men/women with ASD (line 186)?
- For Figure 2, it would be useful to label the 3 distinctive trajectories rather than Avg 1, Avg2, etc. in the figure caption just to be clear. Also, please label x and y axes
- For Figure 3, please label x and y axes in the figure so it is clear at a glance.
Response 5: ASD was more common for men (73%) than women (27%). The exact figures are given in table 1.
We have revised the figures and hope that they are clearer now. We also give explicit descriptions about the abbreviations in the figure titles.
We have also labelled both the Y and X axis in Figure 2, to be in line with the Figure 2: Y axis: probability of unemployment; X-axis: follow-up years from entry year (entry year is marked as 0 and the there is a four year follow-up after, altogether 5 year follow-up for unemployment.)
Point 6: Discussion:
- The discussion is appropriate and informational.
- My only suggestion would be to consider the impact of co-occurring conditions on unemployment in those with ASD. Anxiety and depression are mentioned in the Introduction but not mentioned again. Many individuals with ASD have co-occurring anxiety disorders which can limit suitability for work. Further, physical symptoms like gastrointestinal disorders can limit one’s ability to work, and they are quite common in ASD. Perhaps these could be considered for future studies - to highlight the importance of treating co-occurring conditions if they impact employment in those with ASD.
Response 6: Thank you very much for your comments and suggestion. We have added a reference to our parallel study, which uses the same data and addresses e.g. the important question of co-morbidity. We have also revised the conclusion to mention this point. In addition to ASD, the study focused on other key neurodevelopmental disorders. The conclusion of that recent study was “Early-onset neurodevelopmental disorders, particularly with comorbidity, have a far-reaching impact on adult life in terms of sickness absence and disability pension”. Therefore, the comorbid conditions are not covered here, and including them would have made the study quite complex.
Ref 25: Virtanen M, Lallukka T, Kivimäki M, Alexanderson K, Ervasti J, Mittendorfer-Rutz E. Neurodevelopmental disorders among young adults and the risk of sickness absence and disability pension: a nationwide register linkage study. Scand J Work Environ Health 2020, [Feb 20 Epub ahead of print], 1-8.
Point 7: Conclusions
- The conclutions are appropriate. I have no further comment.
Response 7: Thank you.
Overall, this is a very well written manuscript and equally well conducted study. I do not have many comments. I congratulate the authors on their important work and wish them well with their future studies to help those with ASD.
Response: Thank you, we appreciate your very positive and constructive comments.
